# Permafrost Effect on the Spatial Distribution of $CO_2$ Emission in the North of Western Siberia (Russia)

Olga Goncharova [1,*] , Georgy Matyshak [1], Maria Timofeeva [1,2], Stanislav Chuvanov [1,2], Matvey Tarkhov [1] and Anna Isaeva [1,3]

1   Soil Science Department, Lomonosov Moscow State University, Moscow 119991, Russia
2   Dokuchaev Soil Science Institute, Moscow 119017, Russia
3   Yu. A. Izrael Institute of Global Climate and Ecology (IGCE), Moscow 107258, Russia
*   Correspondence: goncholgaj@gmail.com

**Abstract:** The landscapes in the discontinuous permafrost area of Western Siberia are unique objects for assessing the direct and indirect impact of permafrost on greenhouse gas fluxes. The aim of this study was to identify the influence of permafrost on the $CO_2$ emission at the landscape and local levels. The $CO_2$ emission from the soil surface with the removed vegetation cover was measured by the closed chamber method, with simultaneous measurements of topsoil temperature and moisture and thawing depth in forest, palsa, and bog ecosystems in August 2022. The $CO_2$ emissions from the soils of the forest ecosystems averaged 485 mg $CO_2$ m$^{-2}$ h$^{-1}$ and was 3–3.5 times higher than those from the peat soils of the palsa mound and adjacent bog (on average, 150 mg $CO_2$ m$^{-2}$ h$^{-1}$). The high $CO_2$ emission in the forest was due to the mild soil temperature regime, high root biomass, and good water–air permeability of soils in the absence of permafrost. A considerable warming of bog soils, and the redistribution of $CO_2$ between the elevated palsa and the bog depression with water flows above the permafrost table, equalized the values of $CO_2$ emissions from the palsa and bog soils. Soil moisture was a significant factor of the spatial variability in the $CO_2$ emission at all levels. The temperature affected the $CO_2$ emission only at the sites with a shallow thawing depth.

**Keywords:** climate change; greenhouse gases; permafrost table; peatland; chamber method; podzols; cryosols; histosols

## 1. Introduction

It is an undeniable fact that the warming of the Arctic regions is accelerating faster than the typical rate for the planet. This phenomenon is called "Arctic amplification" [1]. According to the latest calculations, arctic has been warming four times faster than the globe [2]. The consequences of this may be not only the widely discussed loss of sea ice, but also widespread terrestrial warming. Warming in the Arctic permafrost zone accelerates the permafrost degradation with an increase in soil temperature and in the active layer thickness [3]. Both phenomena lead to the activation of the mineralization of organic matter stored in soils and in the upper permafrost [4,5].

It is significant that, occupying only 15% of the global pedosphere, permafrost-affected soils contain about 60% of the global soil carbon (C). Current estimates report 1000 ± 150 PgC in the permafrost-affected soils and uppermost permafrost layer [6,7]. This fact determines the key role of soils in regulating the terrestrial C balance. Temperature changes predicted by model calculations can radically change the existing C balance of the permafrost zone, which, ultimately, will contribute to a significant increase in greenhouse gas releases. As a result of the rise in temperatures and $CO_2$ concentrations, positive feedback in relation to climate change is expected [8–13], that may lead to impeding global efforts in tackling climate change threats. In this sense, a thorough systematic review of existing permafrost carbon cycles is needed to update our understanding of ongoing climate change processes, as well as in supporting climate change mitigation and adaptation

actions [14]. Focusing on countries with mainly resource-based economies and being, at the same time, the highest carbon emitters, is crucial for the development and practical use of science-based mitigation measures and policies [15,16]. Nevertheless, available permafrost-oriented climate change models are sometimes highly uncertain and require further calibration using original data derived from the field.

Modern climate models describing the response of permafrost carbon behavior to global warming indicate moderate C losses from northern ecosystems. The influence of permafrost thawing on the ecosystem carbon balance remains poorly understood, and field observations contradict model predictions [17]. Sometimes, model predictions of C emissions are significantly lower than those expected from field or laboratory experiments [5,18], but the simulation results are unreliable due to the low resolution and uncertainty of the initial model assumptions. Improving the validity of input data and the parameterization of the processes should improve the calibrations and modeling results at a range of scales from local to global [19,20]. The rates of organic matter transformation depend on the physical properties and biogeochemical processes of permafrost-affected soils. Temperature is probably the main driver [21–23]. The positive effect of temperature increase on the decomposition of organic matter and on primary productivity has been widely demonstrated for northern ecosystems [10,24]. In addition, the intensity of $CO_2$ production also depends on soil moisture and groundwater level [25]. These parameters control the redox conditions of the soil and the diffusion of gas in the soil profile. Waterlogging, which induces anaerobic conditions, reduces $CO_2$ emissions but can induce methane release [26].

Western Siberia is a large region in Russia, with mostly flat topography, between the Ural Mountains and the Central Siberian Plateau. This part of Siberia mainly consists of wetlands that occupy approximately 0.7 million km$^2$ [27] or 13% of the total area of mires in the world [28]. The main part of the wetlands is located in the southern part of Western Siberia. The northern part is found in the permafrost zone. Continuous permafrost of great thickness has been mapped in the north of Western Siberia beyond the Arctic Circle. To the south, permafrost is divided into two layers forming a "dovetail" structure in cross section. The surface layer, which is about 50 m in thickness, occurs discontinuously between 66° and 64° N [29]. Everywhere, areas with discontinuous and sporadic permafrost (Sweden, Norway, Russia, Canada, Alaska) are currently experiencing extensive permafrost degradation [30–34]. Obtaining information on the distribution and change in the permafrost situation for each specific region is necessary, not only from a scientific point of view in terms of climate change but also for the purpose of infrastructure planning and commercial activities [31], especially given the resource-producing role of the region.

In the discontinuous permafrost zone of Western Siberia, in the forest–tundra zone, specific landscapes are formed called palsa mires or palsa bogs, which belong to the arctic or "frozen" peatlands [35,36]. Typical tundra areas affected by the influence of close-lying permafrost neighbor extensive woodlands, where permafrost is currently absent. These landscapes are unique objects for assessing the direct and indirect impact of permafrost on greenhouse gas fluxes. The aim of this study was to identify the influence of permafrost on $CO_2$ emissions at the landscape and local levels. We attempted to answer not only the question of how the presence or absence of permafrost affects $CO_2$ emission, but also of how the variability of the thawing depth and permafrost table irregularities affect it.

## 2. Materials and Methods

### 2.1. Study Site

The study area is located in the north of Western Siberia (65°18′52.8″ N, 72°52′54.2″ E), in the discontinuous permafrost zone (Figure 1A). Permafrost occupies > 50% of the territory and is mainly confined to palsa landscapes. Permafrost is absent in the forested areas and wetlands (bogs). The study was conducted within the framework of long-term monitoring research of the Earth Cryosphere Institute and Lomonosov Moscow State University Soil Science Department. Their research is focused on soil, climate, permafrost and plant ecology issues, and contributes to the Circumpolar Active Layer Monitoring (CALM)

program [37–41]. The research site is located on the interfluve of the Kheygiyakha and Levaya Khetta rivers, 40 km from the Nadym city (Tyumen region). The climate of the area is continental, with very cold winters. According to the weather station in Nadym, 30 km north of the key area, the mean annual air temperature since 2004 has been −4.5 °C (with variation from −2.4 to −6.8 °C), and the mean annual precipitation has been 550 mm (variation 466–687 mm).

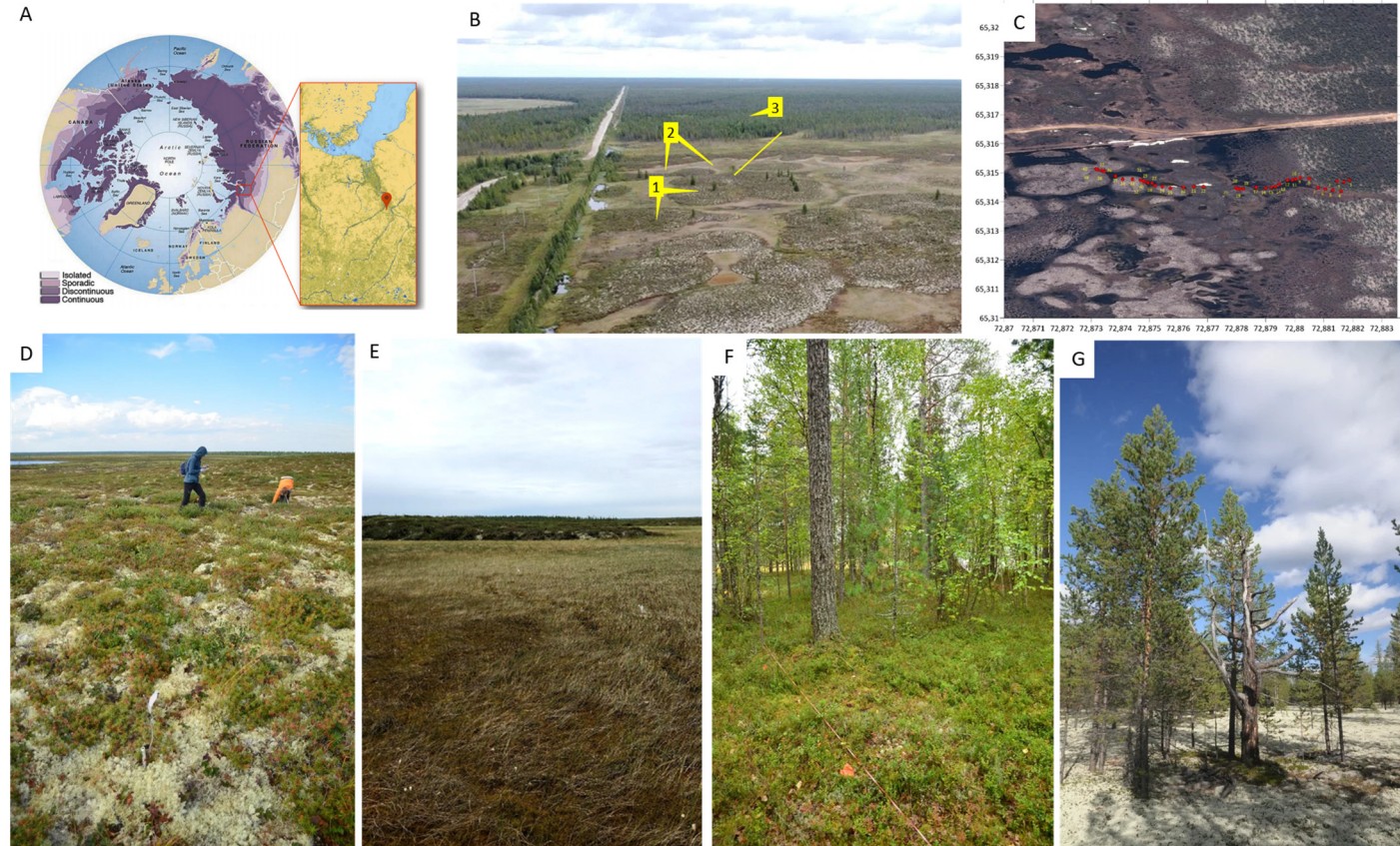

**Figure 1.** (**A**) Permafrost distribution map and location of the study area in the Western Siberia region, Russia. Credit: Map by Philippe Rekacewicz, UNEP/GRIDArendal; data from International Permafrost Association, 1998. Circumpolar Active-Layer Permafrost System (CAPS), version 1.0; (**B**) key objects of research: 1—palsa, 2—bog, 3—forest, yellow line—transect with sampling points (numbers and points); (**C**) location of sampling points on the transect; typical landscapes of the study area: (**D**) palsa (frozen peat plateau), (**E**) ombrotrophic sphagnum peat bog, (**F**) green moss pine forest and (**G**) lichen pine forest.

Studies to assess the influence of the presence and depth of permafrost on $CO_2$ emissions were carried out in three landscapes in August 2022. All sites were located in a small area (several square kilometers) in close proximity to one another (Figure 1B). Forest areas were represented by the green moss and lichen pine stands with a mild hilly-depression relief (absolute height about 22 m asl) without permafrost (Figure 1F,G). The tree layer consisted of *Pinus sibirica*, *Larix sibirica* and *Betula* sp.; species of the Ericaceae family predominate in the shrub layer; and *Polytrichum strictum*, *Cladonia rangiferina* and *Sphagnum* sp. composed the moss-and-lichen layer. Spodosols (podzols) without permafrost [42,43], developing from sandy parent material of the lacustrine–alluvial genesis, predominated in the well-drained forest ecosystems.

The palsa sites were on flat and slightly inclined surfaces with cloudberry–sphagnum cover with participation of *Cladonia rangiferina*, *C. stellaris*, *C. sylvatica*, *Sphagnum* sp., *Betula nana*, *Rubus chamaemorus*, *Ledum* sp., *Vaccinium uliginosum* and *Vaccinium myrtillus*

(Figure 1D). Palsa sites were elevated by 0.5–1.5 m above the adjacent bog due to frost heaving. The active layer consisted of peat horizons. Permafrost was found at an average depth of 60 cm and was represented by sandy loamy or sandy deposits. The third landscape was a waterlogged ombrotrophic sphagnum peat bog (Figure 1E) within a waterlogged depression between palsas (absolute height about 20 m asl). The bog water table level (WTL) was within 0.1–0.25 m of the soil surface. Vegetation consisted of *Sphagnum fuscum*, *Eriophorum vaginatum* and *Carex* sp. Permafrost was not found within 2 m. A heterogeneous complex of gelisols (cryosols) and histosols was described in palsa and bog landscapes [44]. There were 20–130 sampling points in each of the three landscapes (280 points in total).

In addition, sampling points were located along a transect that intersects all landscapes. On the transect, 40 points were studied almost along a straight line (Figure 1C).

*2.2. Field Methods*

The measurements were carried out for 3 weeks in August 2022. Some measurements were made at each sampling point: temperature and moisture of the surface soil layer, $CO_2$ emission from the soil surface, thickness of the active layer (for palsa points) and WTL (for bog points). The $CO_2$ emission from the soil surface with the removed vegetation cover was measured in daytime (from 10 a.m. to 15 p.m.) by the closed chamber method [45]. Opaque chambers were used to assess only soil respiration, including the respiration of roots and soil microorganisms. The chamber method gives information on spatial variability of fluxes in different ecosystems located at short distances in permafrost landscapes [46]. The $CO_2$ concentration in the probes was measured by a portable gas analyzer LiCOR LI-830 (measurement range 0–20,000 ppm, accuracy within 3% of reading). Closed chambers were made of stainless steel and had the volume of 0.9 L. They were placed on special bases and pre-installed on the measurement points with a depth of 2–4 cm after the preliminary removal of vegetation. Gas samples from the chambers were taken with a syringe (20 cm$^3$) through rubber plugs immediately after the insertion of the chamber and 10 min after this. Before sampling, air in the chamber was mixed via triplicate pumping by the plunger of the syringe. Previous studies showed that the gas concentration in the chamber increases linearly during the first 10–15 min after the installation. The flow value was calculated using Equation (1):

$$Q = \frac{\Delta C \times P \times M \times h}{100 \times R \times T \times \Delta t} = \frac{3.18 \times \Delta C \times P \times h}{T \times \Delta t} \tag{1}$$

where Q, $\text{mgCO}_2 \text{ m}^{-2} \text{ h}^{-1}$ is the $CO_2$ emission; P, kPa is the atmosphere pressure; M, kg mol$^{-1}$ is the molar mass; T, °K is the temperature; R, 8.314 J mol$^{-1}$ K$^{-1}$ is the molar gas constant; $\Delta C$, ppm is the change in gas concentration in the chamber; $\Delta t$, min is the exposure time; h, cm is the camera height; and 3.18 is a coefficient that takes into account the values of the constants included in the equation and the ratio of dimensions.

Soil water content (SWC) in the upper 10 cm was measured in triplicate at each point with a FieldScout TDR 100 Soil Moisture Meter (resolution 0.1%, accuracy $\pm$ 3.0%), simultaneously with emission measurements. Soil temperature was measured with an electronic thermometer TP3001 (resolution 0.1 °C, accuracy $\pm$ 0.5 °C). The WTL was measured with a ruler. The thickness of the active layer was measured with a metal rod of 10 mm in diameter and 1.2 m in length (specially made for soil research). All fieldwork on the transect was completed on the same day.

One-way analysis of variance (ANOVA) was used to test the differences in $CO_2$ emissions and environmental factors (soil temperature and SWC) among the different landscapes. Significance was determined using Fisher's least significant difference (LSD) test at a probability level of 95% ($p < 0.05$). The correlations between $CO_2$ emissions and environmental factors were tested using the Pearson correlation analysis and multiple linear regression analyses (significance level $p < 0.05$, $p < 0.1$) in StatSoft Statistica 8.0.

## 3. Results

### 3.1. Topsoil Temperature and SWC

The average temperature of the upper soil horizon at the measurement points was 9.7 °C with variation from 2.0 to 18.3 °C. The coefficient of variation (CV = 36%) was low, taking into account the diversity of studied landscapes. However, the soil temperatures of the palsa, forest and bog differed significantly according to the ANOVA. The average values were 6.6, 10.2 and 13.1 °C, respectively (Figure 2A). The SWC varied very widely: from a few percent in forest soils to more than 70% in bog soils. According to ANOVA, the SWC in soils of different ecosystems differed significantly (Figure 2B).

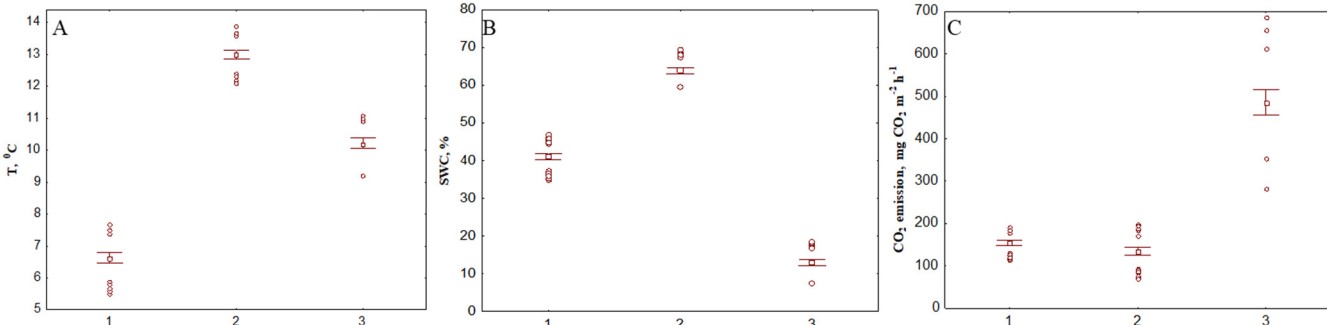

**Figure 2.** Soil (**A**) temperature and (**B**) moisture (SWC) and (**C**) $CO_2$ emission (**C**) in the studied (1) palsa, (2) bog and (3) forest ecosystems. Averages, standard errors of the mean and outliers are shown.

### 3.2. WTL and Thawing Depth

For bog ecosystems, the WTL was estimated. It varied from 0 cm (water on the surface) to 25 cm. Open water was most often observed in the string–flark complex.

In 2022, when the studies were carried out, the maximum depth of seasonal thawing over the past 20 years was observed [47]. In this regard, in many points of the palsa, the permafrost table depth exceeded 1.2 m. The usual thawing depth in the same areas in August is 30–70 cm [40]. The average thawing depth (calculated from the points where permafrost was found (n = 114), the other points were excluded from the calculations) was 84 cm (range from 35 to 116 cm). As mentioned above, the permafrost in the study area is associated only with the palsa.

### 3.3. $CO_2$ Emission

The $CO_2$ emission varied over a very wide range: from 6 to 740 mg $CO_2$ m$^{-2}$ h$^{-1}$, while the CV was 74%. According to this indicator, the palsa and the surrounding bog did not differ significantly from one another; the average rates of $CO_2$ emission were 154 mg $CO_2$ m$^{-2}$ h$^{-1}$ (CV = 44%) and 135 mg $CO_2$ m$^{-2}$ h$^{-1}$ (CV = 73%), respectively (Figure 2C). The $CO_2$ emission was significantly higher in the forest ecosystem with an average value of 485 mg $CO_2$ m$^{-2}$ h$^{-1}$ (CV = 28%) than in the palsa–bog complex. The bog ecosystems were characterized by a very high spatial variability of $CO_2$ emission.

The variability of $CO_2$ emission is well discernible from the data obtained on the transect crossing all the studied landscapes (Figure 3). The length of the transect was about 200 m, with points located at a distance of 2 to 10 m from one another. The largest number of outliers were observed for bog areas. One can note a trend towards an increase in $CO_2$ emissions in bogs immediately adjacent to palsas (points 19, 27, 32, 35).

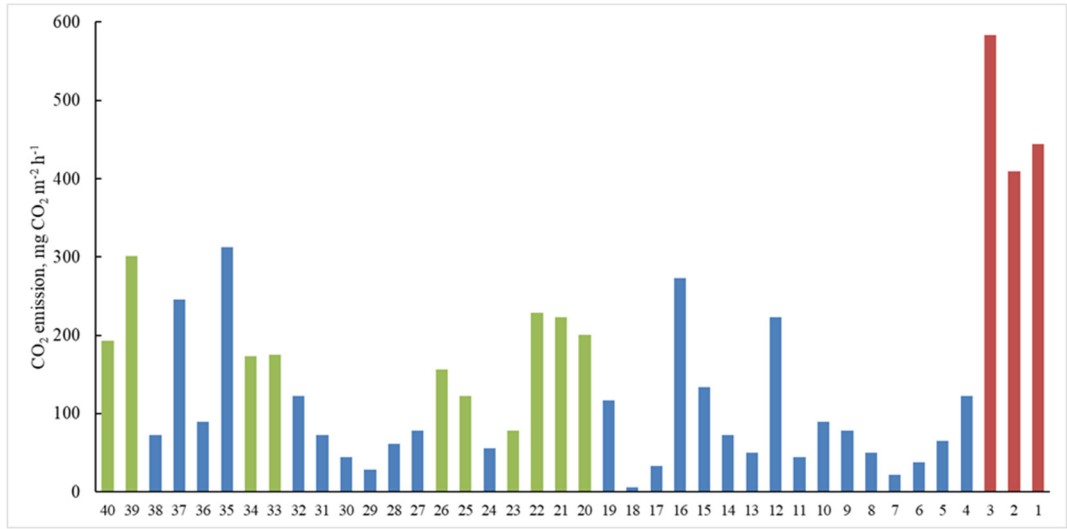

**Figure 3.** $CO_2$ emission on the transect. Red color—forest points, blue color—bog, green color—palsa; the numbers show the sampling points.

### 3.4. Correlation between $CO_2$ Emission and Environmental Factors

According to Pearson's analysis, $CO_2$ emissions (for all 280 points) did not correlate with the temperature of the topsoil and significantly negatively correlated with SWC ($r = -0.37$, $p < 0.05$) (Figure 4). When both factors were included in the analysis for the entire data set (multiple regression), they turned out to be significant, but the explained variance was less than 25%. For individual ecosystems, other patterns were observed. A weak negative correlation between topsoil temperature and $CO_2$ emission was observed for bog ecosystems ($r = -0.22$, $p < 0.05$). Individually, in forest and palsa ecosystems, the $CO_2$ emissions positively correlated with the SWC (for forest ecosystem, $r = 0.60$ at $p < 0.05$; for palsa, $r = 0.16$ at $p < 0.1$) and did not correlate with the temperature. According to the correlation analysis, the thickness of the active layer had no effect on the $CO_2$ emissions from all the studied palsas, but positively correlated with emission from palsas on the transect ($r = 0.60$, $p < 0.1$). In contrast to the total sample, a high positive correlation of temperature and active layer thickness was observed for the transect ($r = 0.83$, $p < 0.05$). It should be noted that only points with the active layer thickness of less than 1.2 m (rod length) were included to the analysis. The WTL also did not have a significant effect on the $CO_2$ emissions from the bog.

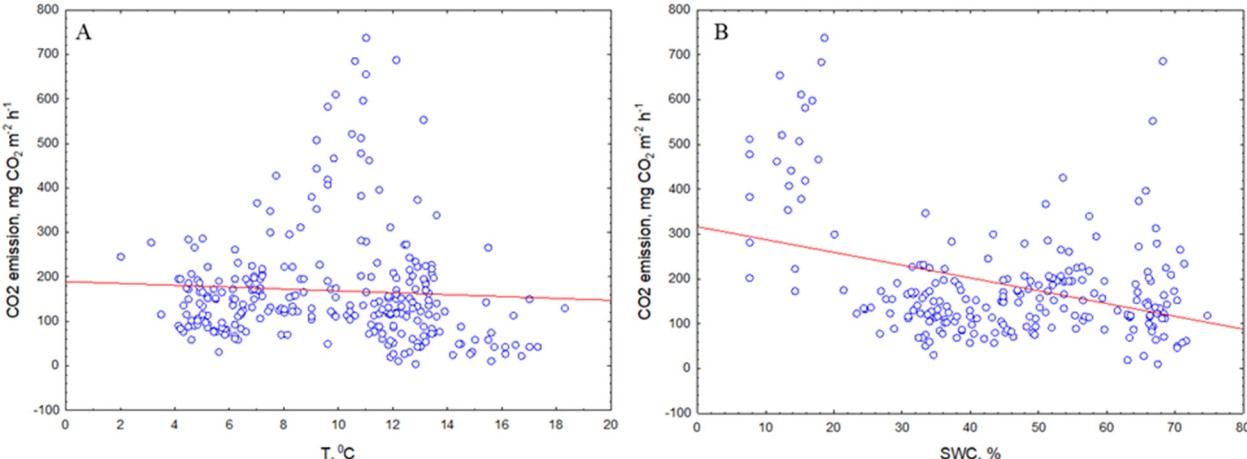

**Figure 4.** Correlation of $CO_2$ emission with topsoil (**A**) temperature and (**B**) moisture (SWC) for all sampling points.

## 4. Discussion

### 4.1. Drivers of Permafrost Impact on $CO_2$ Emissions

We compared our data for key ecosystems with the data of other studies for regions with similar biological and climatic conditions. Similar ecosystems—northern taiga and forest–tundra ecotone with palsa–bog complexes—are found in the European part of Russia, in Fennoscandia and in Alaska. It transpired that most of the data on carbon dioxide fluxes for tundra ecosystems, in particular, for palsas, are data on ecosystem respiration and net ecosystem exchange. Data on soil respiration are extremely scarce. That is, when assessing the resulting flux, the contribution of heterotrophic soil respiration is not evaluated. An insufficiency of data is also pointed out in some reviews [48]. The average values of $CO_2$ emission obtained in our study for the northern taiga ecosystems and palsas (peatlands) are consistent with data presented in other studies [49–52]. Values for bogs are higher than those presented for similar ecosystems.

Forest ecosystems in the study area are characterized by high (3–3.5 times) $CO_2$ emission values compared to adjacent palsa–bog complexes. This is due to a mild soil temperature regime in the absence of permafrost, high root biomass and, as a result, increased root respiration, and favorable water and air permeability of soils. During the observation period, the temperature of forest topsoil was, on average, 4 °C higher than the temperature of palsa topsoil. The supplies of root biomass in forest soils range from 300 to 3000 g m$^{-2}$ depending on the type of plant communities, and the contribution of root respiration to the total soil respiration can reach 80% [53]. Currently, the influence of permafrost on the formation of forest landscapes and soils is minimal. The unique feature of these landscapes is the presence of paleocryogenic phenomena developed in the early Holocene. In the study area, they manifest themselves in specific topographic features, parent materials and vegetation associations [44] (Figure 1G). For example, among typical green moss pine forests, there are areas of lichen pine forests. Soils of lichen associations are characterized by low supplies of root biomass and low $CO_2$ emissions. The high variability of soil respiration in forest ecosystems is also due to these phenomena (Figure 2C). Thus, according to the data of this year, the average emission in the green moss pine forest was 380 mg $CO_2$ m$^{-2}$ h$^{-1}$, and in the lichen forest 520 mg $CO_2$ m$^{-2}$ h$^{-1}$. This fact was also recorded by us in previous long-term studies [53].

Modern cryogenic processes play an important role in the dynamics of subarctic landscapes. One of their manifestations is the development of specific landforms consisting of peat plateaus and mounds with shallow permafrost and ombrotrophic bogs—palsa–bog complexes. Palsas are dynamic formations; their evolution consists of several stages, each of which can be found in the study area [54,55]. The soil cover of the palsa–bog complex is subjected to cyclic development due to the influence of cryogenic processes. This results in significant heterogeneity of the soil cover pattern, soil properties and greenhouse gases fluxes [44]. We obtained data that are not fully consistent with the literature on the role of bog ecosystems in $CO_2$ emissions. In our study, the rate of $CO_2$ emission from bog ecosystems does not differ from that of palsa, and averages about 150 mg $CO_2$ m$^{-2}$ h$^{-1}$. Taking into account the fact that a significant amount of methane is also emitted from the surface of bogs [56], the role of bogs of this type as a carbon sink is questionable. What are the reasons for the high $CO_2$ emissions from the studied bogs? First of all, it is the good warming of bog soils and waters in comparison with other landscapes. According to our data, the bogs are characterized by the maximum temperatures of the upper horizons. This causes both the intensification of microbiological processes and the physical release of gas. It is known that, with an increase in temperature, the solubility of gases decreases and degassing of solutions occurs. The second reason is the hydrological transport of carbon dioxide. The supra-permafrost waters formed in palsas in the course of active layer thawing have subzero temperatures and contain high concentrations of $CO_2$ [56]. These waters flow atop the permafrost table with uneven topography into the surrounding bogs, thus saturating the bog water with $CO_2$. This mechanism is confirmed by the increased values of $CO_2$ emission in the edge parts of the bogs (Figure 3). Thus, part of the gas that

was formed in the process of soil respiration on the palsa is transferred to bog ecosystems, where it is emitted. We believe that this is the underlying mechanism for the leveling of the $CO_2$ emission in the palsa–bog complex.

The maximum spatial variability of $CO_2$ emission was found for bog ecosystems, which is due to the "edge effect" described above, as well as to the presence of "air pockets" with a high content of biogenic gases, which are typical for bogs [57]. The low variability of hydrothermal conditions in bogs also attests to these reasons.

*4.2. Response of CO$_2$ Emissions to Environmental Factors*

The topsoil temperature was not the driver of the spatial variability of $CO_2$ emissions, either between landscapes or within them. A weak correlation between temperature and emission was only found for bog ecosystems. The fact that the role of soil temperature in the spatial distribution of $CO_2$ emission is not high is also noted by other authors for different ecosystems [50,58]. It should be noted that the active layer thickness in the palsas determined the temperature regime only for soils in areas with a shallow thawing depth (transect). Furthermore, for the transect, a high (but not significant) relationship between $CO_2$ emission and the active layer thickness was revealed. We consider that the lack of correlation between both the topsoil temperature and the active layer thickness, as well as between the active layer thickness and $CO_2$ emissions, is due to the deep soil thawing in 2022. With a thawing depth of more than one meter, the influence of permafrost on the topsoil temperature is negligible. In previous studies, when assessing the influence of thawing depth on the soil temperature regime, a high positive relationship between them was revealed [56].

Based on our results, soil moisture has the maximum impact on the spatial variability of $CO_2$ emission. At the landscape level, an increase in moisture leads to a decrease in $CO_2$ emissions. Thus, the general trend is a decrease in $CO_2$ emissions from automorphic to hydromorphic landscapes. This is consistent with a known trend for boreal and arctic regions [51]. Trends for individual ecosystems are also predictable: moisture has no effect on $CO_2$ emission in bogs and has a positive correlation with $CO_2$ emission in forests and palsas. We suppose that some moisture deficit in the palsas in summer may be due to their good drainage. The elevated position of the palsas relative to the surrounding bogs causes the outflow of both supra-permafrost and surface waters. It is obvious that such a situation will be observed as long as the permafrost table is above the level of bog water.

**5. Conclusions**

Landscape and local-scale studies in the discontinuous permafrost zone of Western Siberia in August 2022 showed that permafrost plays a key role in regulating the carbon cycle of ecosystems, including $CO_2$ emissions from soils. The combination of contemporary and paleocryogenic processes determines the specificity of the formation and functioning of local ecosystems and their high heterogeneity and dynamism. This is also reflected in the specific features of their carbon cycle. The rate of $CO_2$ emission from soils of automorphic forest ecosystems ($485 \pm 135$ mg $CO_2$ m$^{-2}$ h$^{-1}$) is 3–3.5 times higher than that from soils of the palsa–bog complex, which is explained by the favorable hydrothermal regime, high root biomass and good water–air properties of the soils of forest ecosystems. The rates of $CO_2$ emission from palsa ($137 \pm 67$ mg $CO_2$ m$^{-2}$ h$^{-1}$) and bog soils ($123 \pm 101$ mg $CO_2$ m$^{-2}$ h$^{-1}$) are nearly equal despite the significant difference in hydrothermal conditions. The reasons for this are both the multidirectional factors and the physical processes. The presence of permafrost leads to a redistribution of $CO_2$ fluxes. The whole volume of $CO_2$ produced by palsa soils is not released from their surface; some part of $CO_2$ produced in palsa soils is laterally transported with supra-permafrost water flows and is released from the surface of bogs.

Soil moisture is a significant factor affecting spatial variability in $CO_2$ emissions at different levels. At the landscape level, an increase in moisture leads to a decrease in $CO_2$ emissions from automorphic to hydromorphic landscapes. At the local level, an increase

in moisture contributes to an increase in $CO_2$ emissions for nonhydromorphic landscapes. The temperature affects the $CO_2$ emission at the sites with a shallow thawing depth. In turn, surface soil temperatures are correlated with the depth of seasonal thawing.

In further studies, it is planned that we will summarize data on carbon dioxide emissions over several years, as well as estimate methane emissions from the soils of the region. The summarized data can be used to model the dynamics of processes associated with climate change in the Arctic zone and related landscape changes.

**Author Contributions:** Conceptualization, O.G. and G.M.; methodology, O.G. and G.M.; investigation, O.G., G.M., M.T. (Maria Timofeeva), M.T. (Matvey Tarkhov), S.C. and A.I.; writing—original draft preparation, O.G. and G.M.; project administration, G.M. All authors have read and agreed to the published version of the manuscript.

**Funding:** This research was financially supported by the Russian Science Foundation (RSF), grant № 22-24-00408 "Carbon in the soil-water-atmosphere system of associated landscapes of the Western Siberia permafrost zone".

**Data Availability Statement:** The data presented in this study are available on request from the corresponding author. The data is not publicly available because the database for the corresponding year has not yet been registered.

**Conflicts of Interest:** The authors declare no conflict of interest.

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
