# Peer review of "Permafrost Effect on the Spatial Distribution of CO2 Emission in the North of Western Siberia (Russia)"

_carbon, 2022_

Round 1
Reviewer 1 Report
The fact that the climate warming in the arctic regions is much faster that the average for the planet is undeniable. This phenomenon is called “Arctic amplification” and the excess is about four times. Landscape- and local-scale studies in discontinuous permafrost part of Western Siberia in August 2022 showed that permafrost plays a key role in regulating the carbon cycle of ecosystems, also in terms of soil CO2 emission.
The study is to identify the effect of permafrost on CO2 emission at the landscape and local levels. The methodology that was used to examine the effect was a one-way analysis of variance called ANOVA. The results show that the high CO2 emission in the forest is caused by the mild soil temperature regime, high root biomass supply and favorable water-air permeability of soils in the absence of permafrost. Soil moisture was a significant factor of the CO2 emission spatial variability at all of the levels.
Overall, the authors wrote that the paper is very well and detailed. They explained the current situation of the problem/case and the structure of the paper is also appropriate. A point of criticism is that you should add a chapter of literature, so the reader knows what is going on while reading and it is not too overwhelming at first. Refer also to the issue of resource abundance and its effects on emissions and environmental degradation, at least cursorily. Siberia is abundant in natural resources, such as oil and gas, metals and minerals. To this end, refer to
https://www.brookings.edu/articles/the-siberian-curse-does-russias-geography-doom-its-chances-for-market-reform/
https://doi.org/10.1016/j.seps.2020.100936
https://doi.org/10.3390/en13153956
The methodology that is used by the authors should be briefly mentioned in the abstract. They should also explain the rationale for using Pearson`s rho. In addition, the value of the correlation coefficient should be reported. Consider also using a simple OLS instead of the correlation coefficient. Despite that, everything that seemed necessary for the study was explained in the text. The length of the paper is good and all of the calculations are explained. In the Conclusion part of the paper, the authors mentioned everything, that is important for it. Also, an outlook for future research is added in the conclusion, where the authors mention the use of the study for the future.
Author Response
Responses to the Reviewer’s Comments for Manuscript “ID carbon-2306816, Permafrost Effect on the Spatial Distribution of CO2 Emission in the North of West Siberia (Russia)"
Reviewing: 1
The fact that the climate warming in the arctic regions is much faster that the average for the planet is undeniable. This phenomenon is called “Arctic amplification” and the excess is about four times. Landscape- and local-scale studies in discontinuous permafrost part of Western Siberia in August 2022 showed that permafrost plays a key role in regulating the carbon cycle of ecosystems, also in terms of soil CO2 emission.
The study is to identify the effect of permafrost on CO2 emission at the landscape and local levels. The methodology that was used to examine the effect was a one-way analysis of variance called ANOVA. The results show that the high CO2 emission in the forest is caused by the mild soil temperature regime, high root biomass supply and favorable water-air permeability of soils in the absence of permafrost. Soil moisture was a significant factor of the CO2 emission spatial variability at all of the levels.
Overall, the authors wrote that the paper is very well and detailed. They explained the current situation of the problem/case and the structure of the paper is also appropriate. A point of criticism is that you should add a chapter of literature, so the reader knows what is going on while reading and it is not too overwhelming at first. Refer also to the issue of resource abundance and its effects on emissions and environmental degradation, at least cursorily. Siberia is abundant in natural resources, such as oil and gas, metals and minerals. To this end, refer to
https://www.brookings.edu/articles/the-siberian-curse-does-russias-geography-doom-its-chances-for-market-reform/
https://doi.org/10.1016/j.seps.2020.100936
https://doi.org/10.3390/en13153956
Response. The authors thank the reviewer for the work done. We have added the phrase to the literature review. But, we would not like to pay much attention to those aspects that the reviewer proposes. The study area is weakly subject to anthropogenic influence. Our goal was to try to identify the mechanisms of changes in GHG emission, which can be caused by climate changes, both natural and anthropogenically induced. In the literature review, we tried to pay the most attention to the uniqueness of the region where discontinuous permafrost is formed. We proceeded from the features of the special issue in which the article is submitted.
The methodology that is used by the authors should be briefly mentioned in the abstract.
Response. We have described the methodology in the summary (“The CO2 emission from the soil surface with the removed vegetation cover was measured by the closed chamber method with simultaneous measurement of topsoil temperature and moisture, active layer thickness at three landscapes in August 2022”), perhaps you had something else in mind?
They should also explain the rationale for using Pearson`s rho. In addition, the value of the correlation coefficient should be reported. Consider also using a simple OLS instead of the correlation coefficient.
Response. Concerning estimates of correlations. The distribution of values is close to normal, so the Pearson coefficient was used. There was an attempt to use the Spearman rank coefficient, but we did not get a fundamentally different result. Regression analysis was also applied (there is a description in the methods), but high significant correlations were also not identified.
Correlation coefficients have been added to the text.
Despite that, everything that seemed necessary for the study was explained in the text. The length of the paper is good and all of the calculations are explained. In the Conclusion part of the paper, the authors mentioned everything, that is important for it. Also, an outlook for future research is added in the conclusion, where the authors mention the use of the study for the future.
English has been corrected by a professional soil scientist translator.
Reviewer 2 Report
Goncharova et al. “Permafrost Effect on the Spatial Distribution of CO2 Emission in the North of West Siberia (Russia)” investigate summer CO2 fluxes in West Siberian landscapes using chamber measurements. Overall the paper is a relatively simple study with measurements from one local region in August 2022. The work makes some conclusions about the affect of soil temperature and soil water content on CO2 emissions from different landscape types (forest, bog or palsa). In some cases with permafrost is present and active layer depth is also considered. The conclusions are valid at least for the conditions studied, thus the work is worthy of publication in C, with a few revisions. One main point that could have made the study more valuable would have been the simultaneous measurement of CH4 emissions and CO2 emissions, since the authors acknowledge that the complete carbon balance is still uncertain without information on CH4 emissions.
Specific points
Line 50: It would be better to mention not just at the global level, but rather “at a range of scales from local to global”.
Figure 1. Even zoomed in at 300% on my computer screen, panel A is still too small to see clearly.
Figure 2. Labels on these figures are too small to read.
Line 194: “discernibled” should be replaced with “discerned”
Line 211: “no correlated” should be replaced with “not correlated”
Line 242: “But, the specific of these landscapes” should be replaced with “The unique feature of these landscapes”
Line 250: As presented, it looks like the authors are reporting a negative emission (i.e. -520 mg CO2 m2 h-1)
Line 264: As mentioned above, It is unfortunate that the current study only measured CO2 and did not make simultaneous CH4 measurements.
Line 317: “nearly the equal” should be “nearly equal”
Author Response
Responses to the Reviewer’s Comments for Manuscript “ID carbon-2306816, Permafrost Effect on the Spatial Distribution of CO2 Emission in the North of West Siberia (Russia)"
Reviewing: 2
Goncharova et al. “Permafrost Effect on the Spatial Distribution of CO2 Emission in the North of West Siberia (Russia)” investigate summer CO2 fluxes in West Siberian landscapes using chamber measurements. Overall the paper is a relatively simple study with measurements from one local region in August 2022. The work makes some conclusions about the affect of soil temperature and soil water content on CO2 emissions from different landscape types (forest, bog or palsa). In some cases with permafrost is present and active layer depth is also considered. The conclusions are valid at least for the conditions studied, thus the work is worthy of publication in C, with a few revisions. One main point that could have made the study more valuable would have been the simultaneous measurement of CH4 emissions and CO2 emissions, since the authors acknowledge that the complete carbon balance is still uncertain without information on CH4 emissions.
Response:
The authors thank the reviewer for the work done. Regarding the emission of methane. Such measurements are carried out in the study area, but to a much lesser extent than CO2 emission measurements. This is due to the need to transport samples to the laboratory. We did not provide data on methane emissions, because a special issue of the С journal is called “Permafrost and Carbon Dioxide Emission”.
In bogs, methane emission was 11.80±8.82 mg CH4 m-2 h-1 and was almost an order of magnitude higher than from palsa soils (1.09±0.31 mg CH4 m-2 h-1).
Specific points
Line 50: It would be better to mention not just at the global level, but rather “at a range of scales from local to global”.
Response: сhanged
Figure 1. Even zoomed in at 300% on my computer screen, panel A is still too small to see clearly.
Response: Figure 1A shows the position of the study area on the permafrost distribution map. Only schematic. If the reviewer does not mind, I will consult with the editor at the stage of the final submission of the article. Whether it is necessary to increase the picture, or even remove it.
Figure 2. Labels on these figures are too small to read.
Response: We have changed the drawing, but it will probably change in the final version
Line 194: “discernibled” should be replaced with “discerned”
Response: changed
Line 211: “no correlated” should be replaced with “not correlated”
Response: changed
Line 242: “But, the specific of these landscapes” should be replaced with “The unique feature of these landscapes”
Response: changed
Line 250: As presented, it looks like the authors are reporting a negative emission (i.e. -520 mg CO2 m2 h-1)
Response: Removed, it was a dash
Line 264: As mentioned above, It is unfortunate that the current study only measured CO2 and did not make simultaneous CH4 measurements.
Answered above
Line 317: “nearly the equal” should be “nearly equal”
Response: changed
English has been corrected by a professional soil scientist translator.
Round 2
Reviewer 1 Report
The authors have ignored substantial share of my comments. The brief reference to the problems and indicated literature sources of resource abundance has not been understood or deliberately ignored. The authors must follow reviewers' recommendations accordingly.
Author Response
The authors in no way ignored the reviewer's suggestion. Lines (78-81 in file carbon-2306816_rev3_corrections are hidden) were added after the first review. In the latest version, we expanded the overview of the problem (lines 43-51 in file carbon-2306816_rev3_corrections are hidden) and referred to the suggested sources (https://doi.org/10.1016/j.seps.2020.100936 and https://doi.org/10.3390/en13153956). A link was also given to the latest IPCC report. English has been corrected.
Once again, we thank the referee for suggestions for improving the article.
Reviewer 2 Report
The authors response to my review are adequate. The only revision that I would suggest is that they mention methane was also measured and the results could be described in a future paper.
Revised article had possible "track changes" formatting issue. New text was added in red, but old text was not removed, so it became rather unreadable.
Author Response
We thank the referee for his work on improving our manuscript. We added a phrase about methane research in the conclusion chapter. English has been corrected
Round 3
Reviewer 1 Report
The authors did a great job!
Author Response
Thank you. If I understand correctly, there were no comments. English was checked again.